# Mental health stigma and professional help-seeking attitudes a comparison between Cuba and Germany

**Laura Nohr**[1]*, **Alexis Lorenzo Ruiz**[2�], **Juan E. Sandoval Ferrer**[3�], **Ulrike Buhlmann**[1]

**1** Division of Clinical Psychology and Psychotherapy, Department of Psychology and Sport Science, University of Münster, Münster, North Rhine-Westphalia, Germany, **2** Division of Clinical Psychology, Department of Psychology, University of Havana, Ciudad de La Habana, La Habana, Cuba, **3** Division of Psychiatry and Psychology, Outpatient Clinic of Mental Health Service, University Hospital General Calixto García Iñiguez, Ciudad de La Habana, La Habana, Cuba

উ These authors contributed equally to this work.

* nohr@wwu.de

**Data Availability Statement:** Although the authors cannot make their study's data publicly available at the time of publication, all authors commit to make the data underlying the findings described in this

## Abstract

Globally the burden due to mental disorders is continuously increasing. Still, professional help-seeking behavior is not fully understood. To conceive cultural determinants of help-seeking is crucial to reduce personal and social costs of (untreated) mental disorders. The current study investigates mental health stigma and help-seeking attitudes in a Cuban ($n =$ 195) and a German ($n =$ 165) sample. In a questionnaire survey we asked for attitudes towards mental illness and professional help-seeking in the general Cuban and German populations. The cultural context was associated with mental health stigma and professional help-seeking attitudes. Interestingly, Cuban participants reported stronger mental health stigma and more willingness to seek help. In multiple hierarchical regression analyses, community attitudes towards the mentally ill significantly predicted help-seeking attitudes, especially in the Cuban sample. Only in the German sample, more negative individual beliefs about mental illness predicted more self-stigma on help-seeking. Beyond that, cultural context moderated the association between mental health stigma and help-seeking attitudes with a stronger association between the measures in the German sample. However, gender did not predict help-seeking attitudes and self-stigma on help-seeking and no interactions between community attitudes, cultural context, and gender were found in the prediction of help-seeking attitudes. Similarities and differences between the samples are discussed in the light of the cultural contexts and peculiarities of the current samples. Concluding, implications of the current findings are reviewed.

## Introduction

### Mental health stigma

The word stigma is used for (negative) characteristics that tag a person and thus distinguish the affected person from others [1–3]. Precondition of stigmatization is an imbalance of power

study fully available without restriction to those who request the data, in compliance with the PLOS Data Availability policy. For data sets involving personally identifiable information or other sensitive data, data sharing is contingent on the data being handled appropriately by the data requester and in accordance with all applicable local requirements. Unfortunately, the data presented in the current research report cannot be shared publicly. Due to a lack of a specific passage explaining Open Science and public availability of anonymized research data in the informed consent form, the local ethical committee decided to not allow data sharing. When allowing us to use the data for a specific scientific goal, our participants did not consent in a general scientific use. Moreover, our research includes private data of our participants which might allow to identify them when sharing the full underlying data set. Therefore, we are only able to share parts of the data on request and have to verify each request and justify our decision each time. Because an Institutional Data Access Committee does not exist, we would like to invite researchers to contact the corresponding author Prof. Dr. Ulrike Buhlmann (ulrike.buhlmann@wwu.de), the first author Laura Nohr (nohr@wwu.de), or the co-authors (Alexis Lorenzo Ruiz: alexis.lorenzo@psico.uh.cu; Juan E. Sandoval Ferrer: sandoval@infomed.sld.cu) to discuss in detail the research proposal so that we can review the criteria for access to confidential data with the local ethical committee and share data within the scope of the local data protection law that aims to protect our participants. To decide on data requests, we will apply the following criteria: (1) Does the researcher request already processed data (e.g. means, standard deviations, etc.) or raw data? (2) How much data is requested and includes the requested data private data like gender, age, ethnicity, etc. (3) Will the sharing of the requested data violate the informed consent of the participants and thereby the approved protocol of our Institutional Review Board? We guarantee that the raw data, the processed data, and all data analyses scripts are stored and archived at local servers for the long term. Prof. Dr. Ulrike Buhlmann is head of the local work unit. This position is meant to stabilize local scientific infrastructure. Prof. Dr. Ulrike Buhlmann as the corresponding author of the current study report will be responsible for all data requests. Hence, she is host of these servers and provides data storage and data access for at least ten years after publication. Our own aspirations are to guarantee data access for a much longer time to improve scientific practice in the future. Thus, as an additional back-up Laura Nohr will have access to

[2]. In this sense, stigma involves disadvantage and discrimination for the demarcate group. The label *mentally ill* stigmatizes people affected by mental disorder. Stigma is defined as *public stigma* when the community tends to associate mental illness with negative labels and generalizes them [4–6]. Respective labels are generally followed by negative emotional reactions and behavior against affected persons [7–9]. Frequent labels associated with mental illness are dangerous [7, 10], blameworthy [7, 11], unpredictable [11, 12], incompetent [8, 10], dependent [12, 13], difficult-to-treat or not curable [10]. *Self-stigma* applies when an individual accepts the negative stereotypes and identifies with the group of mentally ill persons [9, 14]. Then the person reacts emotionally and behaviorally negative against the self [9, 14–16], for example, in terms of decreased self-esteem, self-efficacy, and hope [17]. Altogether, mental health stigma carries a high burden on affected persons resulting in serious personal consequences [e.g., 4, 6, 18–21] through to higher suicidality [22].

Given the high prevalence of mental illness worldwide [23–26], many persons are affected by mental health stigma. Mental illness is one of the leading causes of disability and disease burden globally [25, 27, 28] and causes high social costs [6, 25, 29, 30]. Unfortunately, the duration of untreated mental illness is associated with adverse clinical outcomes, social costs, and lower quality of life [31–34]. However, a substantial treatment gap exists provided that about two thirds of persons in need do not receive adequate treatment [23, 34, 35], even in high-developed regions like Europe or the USA [34–37]. Although the mental health care gap is well known, treatment barriers remain and efficient strategies to reduce their impact are still lacking. Mental health stigma is one powerful mental health care barrier [e.g., 38–40] which prevents mentally ill persons from seeking professional help.

## Mental health stigma and help-seeking

Professional help-seeking behavior and treatment barriers have been studied in different settings [e.g., 41–47]. Latest systematic reviews indicate at least moderate associations between mental health stigma and professional help-seeking [38]. However, not only *being* mentally ill carries high stigmatization but also *seeking* (professional) help, which also prevents people from accessing treatment [48–50]. Stigma and help-seeking are conceptualized as at least partly culture-bound and embedded in the specific cultural context of the individual and its community [51–54]. Still, most findings come from Western, high-income countries [38]. Accordingly, the influence of culture on mental health stigma and help-seeking behavior should be tested for more diverse cultural groups. Cross-cultural research benefits our understanding of relationships between aspects of mental health stigma, help-seeking behavior, and its interactions with aspects of the social context like personal orientation, gender roles, and exchange rules [52, 53]. Especially interesting cultural contexts might be Latin American cultures. Most research investigates Latin American immigrants in the USA. Literature shows diverse, partly culture-specific mental health care barriers in the community of US-Latin Americans [55]. The mental health stigma seems to be particularly strong [55, 56] while mental health care use seems to be comparably low in US-born and especially immigrant Latin Americans [57]. Compared to other ethnic groups, female US-Latin Americans were more likely to report mental health stigma as a mental health care barrier [58, 59]. Mental health care providers describe the cultural value of *familismo* and the female gender role *marianismo* as important mental health care barriers for female US-Latin Americans [60]. At the same time, Ojeda and Bergstresser [44] found no association in an US-Latin American sample and Rojas-Vilches et al. [61] found stronger associations between mental health stigma and negative help-seeking attitudes for US-Puerto-Ricans and US-Cubans compared to their children.

the data as well and will guaranty data storage within the scope of local data protection laws.

**Funding:** This study was funded by internal financial resources of the University of Münster. The research stay of the first author (LN) in Havana, Cuba, was additionally funded by the travel grant IP@WWU of the University of Münster. The funders were neither involved in the collection, analysis, and interpretation of the data, nor in the design and writing of the study. We acknowledge support from the Open Access Publication Fund of the University of Münster.

**Competing interests:** The authors have declared that no competing interests exist.

The existing evidence from Latin American countries is also mixed. One study conducted in Chile identified a considerable mental health care gap and mental health stigma as one indirect barrier of mental health care use [62]. Caqueo-Urízar et al. [63] found differences in alienation and social withdrawal between Bolivia, Chile, and Peru. Mascayano et al. [64, 65] describe culture-specific findings regarding stigma, the importance of family, and gender roles in Latin American culture. At the same time, Des Courtis et al. [66] and Mileva et al. [67] report stronger mental health stigma in Western compared to Latin American or Caribbean countries. Summarized, the existing (cross-cultural) evidence from Latin America is inconsistent, calling for more research in this field of studies. Therefore, the current study aims to investigate the influence of mental health stigma on professional help-seeking attitudes in two cultural contexts, namely in Cuba and Germany. Thereby, we intend a knowledge gain on a previously understudied Latin American country and add evidence on similarities and differences between these cultures. Above all, a profound understanding on basal relationships in different social contexts will allow us to develop future interventions to reduce mental health stigma and facilitate access to mental health care for diverse populations in the globalized world.

## Cultural contexts of the present study: Cuba and Germany

Cuba is a Caribbean island with most of the population (11.3 million citizens) living in urban areas [77%; 68]. Cuba defines itself as a one-party democracy. Since the Cuban Revolution in 1959 the Cuban Communist Party guides the state [69, 70]. Cuba was classified as a 'high human development' country in the Human Development Report 2019 [68]. In fact, Cuba scores high in the human development indicators *quality of health* and *education*. Still, the gross national income per capita and the overall standard of living are low compared to other high developed countries. One political goal of the socialist government is social justice [71]. In the Public Health Law from 1983 [72, 73] health care is defined as a human right and public responsibility. Hence, health care is free of charge for Cuban citizens and all health care facilities are government owned. Its first priorities are prevention and population health [74]. To reach this, health care is highly structured, community- and family-based. The public health care creates high costs and Cuban government pays low salaries to the health care professionals. Therefore, many health care professionals earn extra money by trading or black labor in the private sector [71]. Further, Cuba sends its health care professionals abroad as part of a paid medical service to steady its economy [73] and without charge to poor countries as part of a sustainable program [75, 76]. Therefore, the Cuban health care system is struggling with shortages in staff and the quality of medical care has diminished [71, 77]. As part of Cuban public health care system, mental health care is also community-based and guided by the *Mental Health Action Plan 2013–2020* [78]. Community Centers of Mental Health offer prevention programs, psychosocial rehabilitation, and general mental health support. Within the social psychiatric approach, they cooperate directly with the communities promoting (psychological) health, improving social relations, and integrating the neighborhood residents and families in the treatment process [76]. For more severe cases of mental illness, psychiatric hospitals provide short-term crisis intervention, short-term hospital treatment, and day hospitals. Besides its mental health care system, Cuban culture is another reason to study mental health stigma and help-seeking in Cuba. Historically, Spanish colonization and African slavery shaped Cuban culture strongly. Cubans identify as Latino, the official language is Spanish [69]. Thus, important cultural values like familismo and Latin American gender roles also apply widely for Cubans. Cuban culture is described as collectivistic or socio-centric [56, 79, 80]. This

means that individuals experience themselves in relation to the social environment which influence individual's cognitive, emotional, and behavioral states [81].

Summarized, Cuba offers a singular opportunity to study culturally informed psychological phenomenon. Beyond that, it is especially interesting to compare Cuban and German culture. Before the breakdown of the Soviet Union and the German Democratic Republic in 1989/90, a strong economic and ideological relation between Cuba and the German Democratic Republic existed. Economic, academic, and occupational exchange programs strengthened the relationship between both countries. Nowadays, Germany forms part of the European Union. It is the country with most inhabitants in the European Union [82.5 million; 82]. Germany defines itself as a democratic and social federal state [83]. Two world wars and the German division have shaped German history and culture. Due to different immigration waves [82, 84], many cultures and religions live together in Germany nowadays. Still, Germany forms part of the Western world and is meant to be a predominantly individualistic culture. Thus, independence, individual choices, and decisions are relatively important in German culture [85, 86]. In the German social state, health insurance is mandatory and the state guaranties health insurance for every inhabitant [87]. Health care and health insurance are based on a solidarity principle, that is all public and private health insurances cover mental health care. To be more precise, mental health care in Germany turned to a community-based strategy in the 1970s and is nowadays mostly provided by private, volunteer, and other organizations [88]. The 16 federal states are responsible for the public mental health care leading to respective regional differences. Besides, there exists a wide gap between inpatient and outpatient services who are provided by different teams and funded separately [88].

Taken together, Cuba and Germany with their quite distinctive characteristics offer an appropriate context to study mental health stigma and help-seeking behavior cross-culturally as they allow us to investigate both universal aspects and culturally informed peculiarities. Further, studying public stigma cross-culturally helps us to comprehend the social circumstances persons live in and have to deal with when recognizing mental health issues and mental health care need. Based on the literature, the current study aims to test the following hypotheses: (1) In both cultural contexts, we suppose higher mental health stigma to decrease the willingness to seek professional help [38, 55]. (2) We expect higher mental health stigma and less willingness to seek professional help in the Cuban (vs. German) sample [55, 56]. (3) We aim to test exploratorily interactions between mental health stigma, gender, and cultural context in the prediction of professional help-seeking attitudes [44, 56].

## Methods

### Sampling

We aimed to sample the general population using convenient sampling in both cultural contexts. We excluded mental health professionals from the study since we expected them to have biased attitudes towards mental illness and professional treatment compared to the general lay population [38, 89, 90]. With respect to a priori power analyses [G*Power 3.1; 91], we aimed to recruit at least a sample size of $n = 160$ in each cultural context to reach an $\alpha$-error of .05 and a sensitivity of $1-\beta = .80$. In the Cuban sample ($n = 195$), a paper-pencil-version of the survey was used. As common in Cuba, the local ethical committee permitted sampling only in limited public spaces. Thus, sampling took place from March to April 2018 in the waiting areas of several medical departments of the university hospital *General Calixto García Iñiguez* in Havana (aesthetic and general surgery, allergology, coloproctology, dermatology, endocrinology, gastroenterology, geriatric medicine, internal medicine). Since the hospital is located in the heart of the Cuban capital, sampling approached mainly citizens of Havana. Still, many

Cubans from other provinces and the country side live in the capital [92]. Besides, the university hospital provides medical care for the whole Cuban population which enabled us to collect data of at least some residents from all over the country. Every patient and her/ his company entering the waiting area were invited to participate voluntarily. As it is common in Cuban culture to accompany relatives to see a doctor, we sampled mainly healthy companions and least affected patients from the waiting area. We did not sample patients of the psychiatric department of the hospital.

In the Cuban sample, data exhibits substantial missing data ranging from 1.5% to 24.1% at item level. To test the underlying missing data mechanism, homogeneity of means and covariances was tested using univariate $t$-test comparisons [93] and Little's MCAR test [94] was applied, $X^2 = 7929.599$, $DF = 7562$, $p = .002$, both indicating that data was not missing completely at random. Ensuing correlation analyses and univariate $t$-test comparisons alludes more to a missing at random missing mechanism than to a missing not at random missing mechanism [93]. Pre-analyses were used to identify auxiliary variables for the imputation models as well as to improve statistical power and reduce potential bias without altering the interpretation of the resulting parameters [93, 95]. Item-level multiple imputation and a fully conditional specification were applied [95]. Due to the high number of variables in the imputation model, we applied parcel summary multiple imputation to achieve satisfying convergence and stable imputations [96, 97]. Graphical diagnostics and the potential scale reduction factor were used for convergence diagnostics as well as to identify an adequate burn-in interval and number of between-imputation iterations. We generated 20 sets of imputations by saving a data set after a burn-in interval of 5000 computational cycles and separated each data set by 2300 iterations. Statistical analyses were applied on each filled-in data set separately and estimated parameters were pooled into a single set of results using Rubin's formulas [98]. Multiple imputation was conducted using the software Blimp [99].

The German sample ($n = 165$) was recruited online by using Inquisit Web 4 [100] and EFS-Survey [101]. The survey link was spread via social media platforms, in newspapers, and in public spaces like bus stations and supermarkets. While we addressed the population of Münster via newspaper advertisement, posters, and flyers in public spaces, we reached out to the broader German population via Internet. We used open and closed groups on social media platforms with titles like *For sell*, *for free*, *to switch* to sample preferably unbiased participants from diverse population segments. Further, we posted our research on the bulletin boards for psychological research where interested persons can participate voluntarily (e.g., Psychologie heute). To protect the privacy of the participants, we did not collect information beyond the scope of the research questions (e.g., their current location). Thus, we cannot control whether we sampled rural and urban population likewise, for example. Data collection took place from June to September 2016. The German dataset shows no missing data because of forced choice as default option in the online survey.

## Measures

Initially, the cross-sectional questionnaire survey acquired sociodemographic information. We asked with single items for age, gender, education, and familiarity with mental illness. To estimate the familiarity with mental illness as an important control variable [102], we asked for previous contact with mentally ill persons on a five-point Likert-scale ranging from 1 (*never*) to 5 (*very often*). To ask for prior professional help-seeking we used the second part of the general help-seeking questionnaire [GHSQ; 103]. If participants answered the initial item affirmative, we further asked for type, frequency, and perceived helpfulness of the received treatment. The questionnaires used to answer the present research questions were presented in the order

described below. Further questionnaires included in the survey and not relevant for the current study will be presented elsewhere. If not explicitly indicated otherwise, original (English) versions of the questionnaires were translated-back translated [104] in German and Cuban Spanish and adapted to the everyday language of the respective cultural context by bilingual and trilingual members of the research team.

**Beliefs towards mental illness scale.** To assess stereotypes about mental illness on an individual level, we used the 21 items of the Beliefs Toward Mental Illness Scale [BMI; 10]. Participants answered each item on a six-point Likert-scale ranging from 0 (*completely disagree*) to 5 (*completely agree*). Items in the BMI are added to a sum score with higher scores indicating more negative stereotypes against mental illness. The BMI has been used in a wide range of cultural contexts and in different languages [105]. Generally, the BMI shows good psychometric properties in different contexts with an internal consistency ranging from $\alpha = .89$ to $\alpha = .92$ [10, 106, 107]. In the current study, BMI shows also good internal consistency in the Cuban sample ($\alpha = .81$) and in the German sample ($\alpha = .90$). The BMI has been used with US-Cuban participants previously [61]. Originally, the authors postulated three subscales named Dangerousness, Poor interpersonal and social skills, and Incurability. Still, the measurement invariance and factor structure remain vague. Hirai et al. [105] postulate a four-factor structure for Latin American samples with a fourth factor representing Embarrassment. Beyond, Royal and Thompson [108] postulate one-dimensionality for a white US-American sample. No previous research about measurement equivalence exists for the German context. Due to this lack of proof, we used the sum score of the scale as suggested by Hirai et al. [105].

**Community attitudes toward mentally ill scale.** Community attitudes towards persons with mental illness were measured by using the 40 items of the Community Attitudes Toward the Mentally Ill Scales [CAMI; 109, German version; 110]. Participants indicate their (dis) agreement to the items on a five-point Likert-scale ranging from 1 (*strongly agree*) to 5 (*strongly disagree*). Lower sum scores indicate more negative community attitudes. The CAMI has already been tested for methodological quality in different cultural contexts [110, 111]. In the present study, the CAMI shows satisfying internal consistencies of $\alpha = .78$ and $\alpha = .94$ in the Cuban and German samples, respectively. The authors postulated four subscales: Authoritarianism, Benevolence, Social restrictiveness, and Community mental health ideology. This theoretical structure was replicated in a German sample [110]. A validation of cultural invariance of a Spanish version in the Latin American context is still pending. Therefore, we conducted the analyses with the sum score.

**Attitudes toward seeking professional psychological help scale.** To capture attitudes towards professional help-seeking, we used the short form of the Attitudes Toward Seeking Professional Psychological Help Scale [ATSPPHS-SF; 112, German version; 113]. Participants had to indicate their (dis)agreement to 10 statements on a four-point Likert scale ranging from 1 (*disagree*) to 4 (*agree*). A sum score was calculated whereby higher values indicate more positive attitudes. We used attitudes as a predictor for future behavior as supported by the Theory of Planned Behavior [114]. Meta-analyses already proofed that self-reported attitudes and help-seeking intentions are good predictors of future health behavior [e.g., 115, 116]. In the current study, internal validity was only moderate in the Cuban ($\alpha = .65$) and the German sample ($\alpha = .79$). Fischer and Farina [112] postulate one general factor of help-seeking attitudes for the ATSPPHS-SF. However, results across different cultural contexts are mixed. Several studies found a two-factor solution [117–120], while others found a three-factor [121] or none adequate factor model [122]. Again, we used the sum score since methodical research for the Cuban and German context is lacking.

**Self-stigma of seeking help scale.** We asked participants for expected cognitive and emotional changes towards themselves in response to professional help-seeking using the 10 items

of the Self-Stigma of Seeking Help scale [SSOSH; 49]. Participants indicate their agreement on a five-point Likert-scale ranging from 1 (*not at all*) to 5 (*extremely well*). Items are summed up to a total sum score whereby higher values indicate more self-stigma of seeking help. Internal validity in the German context has been proofed [123] and was found in the current German sample ($\alpha$ = .89) while it was moderate in the Cuban sample ($\alpha$ = .62). However, these results are only sparsely inferior to the results found by Vogel et al. [49]. The authors suppose a one-factor structure for the questionnaire which was mainly confirmed cross culturally [124] or at least partly replicated [117]. Cross cultural invariance was proofed for a German sample [125] while it is still lacking for Latin American samples. Thus, only the sum score was used for analyses.

Besides descriptive analyses, univariate *t*-tests and multiple hierarchical regressions where applied for the main analyses. Statistical assumptions were examined as suggested by Field et al. [126]. No severe violations were found. Based on theoretical reflections, predictors were entered into the regression model in three sequential blocks. To check for cultural differences, separate regression analyses were applied in the Cuban and German samples. Moderation analyses were used to test two-way and three-way interactions of mental health stigma, cultural context, and gender in the prediction of help-seeking attitudes. Statistical analyses were conducted using the software program R with the mice package [127] and mitml package [128].

## Ethical considerations

To conform with the ethical standards and laws of both cultural contexts, the research study was approved by the Institutional Review Board at the Department of Psychology and Sport Science of the University of Münster and the scientific committee at the Faculty of Psychology of the University of Havana. All participants were informed about the content, goals, conditions, and confidentiality of the research project in written form. Further, they were informed about the voluntary nature of their participation, that no beneficence or maleficence are expected from participation, that data will be stored safely and anonymously, and that they can withdraw from participation without any drawback. Additionally, Cubans were informed verbally and had the chance to ask LN or the Cuban staff any doubt prior to participation which they used frequently. Moreover, contact information of LN and ALR were offered in any case of emergency after participation. German participants were invited to contact LN via mail or telephone to discuss any question which only few participants did. All participants provided written informed consent prior to participation. If Cuban participants withdrew participation afterwards, they kept their questionnaire and were handed in their consent form. German participants had to repeat their consent in the end of the survey. If they did not agree a second time, data was deleted and not included for analysis.

## Results

### Descriptive statistics and group differences

Descriptive analysis, univariate *t*-tests, and $X^2$-tests show substantial differences between the Cuban and German sample. Due to multiple comparisons and alpha error inflation we corrected the alpha-level using the Bonferroni correction as a conservative correction method [126]. Thus, we apply $p < .005$ as criterion for significance and asterisk it. On average, Cubans were older ($M$ = 47.44 years, $SD$ = 15.67 years) than Germans ($M$ = 29.49 years, $SD$ = 10.14 years), $t(337.031)$ = -12.424, $p < .001^*$. Cubans reported on average a minor level of education (51.79%) than Germans (92.12% high school education or higher), $F(1, 893214.5)$ = 69.259, $p < .001^*$. Data show a significant difference in familiarity with mental illness with Cubans reporting less familiarity ($M$ = 1.82, $SD$ = 1.22) than Germans ($M$ = 2.38, $SD$ = 0.97),

$t(347.181) = 4.740$, $p < .001^*$. There was no significant difference in the gender distribution between the Cuban (69.74%) and German sample (70.30% female), $F(1, 187.67) = -0.023$, $p > .99$. Further, analyses show neither a significant difference in previous help-seeking between Cubans (44.10%) and Germans (41.81% received professional help previously), $F(1, 3368.16) = 0.194$, $p = .660$ nor in the perceived helpfulness of the received treatment in the Cuban ($M = 4.04$, $SD = 1.67$) compared to the German sample ($M = 3.93$, $SD = 1.20$), $t(120.048) = 0.469$, $p = .640$. Results show significant differences in mental health stigma. Cubans reported more negative beliefs about mental illness ($M = 47.34$, $SD = 16.57$) than Germans ($M = 36.93$, $SD = 14.10$), $t(316.608) = -6.170$, $p < .001^*$, and analogously Cubans reported more negative community attitudes towards mentally ill persons ($M = 147.36$, $SD = 18.39$) compared to the Germans ($M = 168.01$, $SD = 18.79$), $t(277.206) = 10.018$, $p < .001^*$. Interestingly, Cubans reported at the same time more positive attitudes towards seeking professional help ($M = 22.75$, $SD = 5.12$) compared to Germans ($M = 20.23$, $SD = 5.37$), $t(306.905) = -4.398$, $p < .001^*$ and we found less self-stigma on help-seeking in the Cuban ($M = 19.86$, $SD = 7.34$) than in the German sample ($M = 24.10$, $SD = 8.01$), $t(230.927) = 4.845$, $p < .001^*$).

## Mental health stigma and sociodemographic variables

The overall regression model included in the first step all possibly relevant sociodemographic and control variables. Higher age predicted more positive help-seeking attitudes and less self-stigma on help-seeking. Further, more positive help-seeking attitudes were predicted by higher familiarity with mental illness. In a second step, we entered both stigma measures. More positive community attitudes predicted more positive help-seeking attitudes and less self-stigma. From the first step, only age remained a significant predictor of help-seeking attitudes. In the last step, we added the cultural context to the regression model which predicted significantly differences in the outcome variables. The Cuban cultural context was associated with more positive help-seeking attitudes and less self-stigma. Adding the stigma measures to the model improved the prediction of help-seeking attitudes, $F(243, 2) = 6.547$, $p = .002$, and self-stigma, $F(245, 2) = 10.949$, $p < .001$. By adding the cultural context to the model, the prediction improved again for help-seeking attitudes, $F(238, 1) = 22.335$, $p < .001$, and self-stigma, $F(203, 1) = 41.366$, $p < .001$. The complete regression model explained 20% of variance in the help-seeking attitudes and 25% of variance in self-stigma (Table 1).

In the Cuban sample, higher age and no previous treatment experience predicted more positive attitudes towards professional help-seeking in the first step. This prediction did not change when entering both stigma measures into the model. However, only community attitudes were a significant predictor of help-seeking attitudes and adding the stigma measures did not improve the regression model significantly, $F(2, 180) = 2.368$, $p = .098$, n.s. 15% of the total variance of help-seeking attitudes were explained by the included predictors. In the prediction of self-stigma on help-seeking, community attitudes were the only significant predictor in the Cuban sample. Adding the stigma measures to the regression model improved its prediction significantly, $F(2, 180) = 8.909$, $p < .001$. In total, the model explained 23% of the variance of self-stigma in the Cuban sample (Table 2).

In the German sample, higher familiarity with mental illness predicted more positive help-seeking attitudes significantly when including the sociodemographic predictors alone and after including the stigma measures. Out of the two stigma measures, only more positive community attitudes predicted more positive help-seeking attitudes. In general, the prediction improved by adding the stigma measures, $F(2, 155) = 13.411$, $p < .001$, and a total of 32% of the variance were explained by the model. Regarding the self-stigma on help-seeking, older age and higher familiarity predicted less self-stigma in the first step. After including the stigma

**Table 1. Multiple hierarchical regressions with three blocks of predictors on both help-seeking measures in the complete sample.**

| | N = 360 | ATSPPHS-SF | | | SSOSH | | |
|---|---|---|---|---|---|---|---|
| | | b (95% CI) | SE B | p | b (95% CI) | SE B | p |
| **BLOCK 1** | DEMOGRAPHICS | | | | | | |
| | **Constant** | **18.156 (5.572–30.741)** | **6.218** | **.006**\*\* | **26.064 (9.219–42.909))** | **8.391** | **.003**\*\* |
| | **Age** | **0.091 (0.049–0.132)** | **0.021** | **<.001**\*\*\* | **-0.089 (-0.152 –-0.025)** | **0.032** | **.007**\*\* |
| | Gender (female = 0, male = 1) | -0.721 (-2.048–0.607) | 0.673 | .285 | 0.396 (-1.560–2.352) | 0.992 | .690 |
| | Education (no = 0 / yes = 1) | | | | | | |
| | Primary school | -2.612 (-18.653–13.428) | 8.073 | .747 | 2.329 (-19.986–24.644) | 11.279 | .837 |
| | Secondary school | -1.658 (-13.980–10.664) | 6.091 | .787 | 1.473 (-14.972–17.918) | 8.198 | .858 |
| | Vocational training | -1.346 (-13.833–11.141) | 6.158 | .828 | 0.790 (-15.826–17.407) | 8.265 | .924 |
| | Highschool education | -2.408 (-15.389–10.573) | 6.436 | .710 | 3.042 (-14.191–20.274) | 8.618 | .725 |
| | University degree | -1.282 (-13.745–11.181) | 6.149 | .836 | 1.389 (-15.281–18.060) | 8.293 | .868 |
| | Master | -1.174 (-13.647–11.299) | 6.152 | .850 | 0.548 (-15.996–17.093) | 8.234 | .947 |
| | PhD | -1.033 (-13.501–11.434) | 6.182 | .868 | -1.300 (-17.979–15.378) | 8.338 | .877 |
| | Other | 0.991 (-12.551–14.533) | 6.757 | .884 | -8.611 (-26.701–9.479) | 9.099 | .347 |
| | **Familiarity** | **0.677 (0.170–1.183)** | **0.258** | **.009**\*\* | -0.619 (-1.436–0.197) | **0.414** | **.136** |
| | Previous treatment (no = 0 / yes = 1) | -0.009 (-1.195–1.176) | 0.602 | .988 | -1.098 (-2.872–0.677) | 0.902 | .224 |
| **BLOCK 2** | DEMOGRAPHICS | | | | | | |
| | Constant | 10.352 (-3.799–24.503) | 7.034 | .148 | **41.090 (22.211–59.968)** | **9.451** | **<.001**\*\*\* |
| | **Age** | **0.106 (0.064–0.149)** | **0.022** | **<.001**\*\*\* | **-0.119 (-0.182 –-0.056)** | **0.032** | **<.001**\*\*\* |
| | Gender (female = 0, male = 1) | -0.504 (-1.812–0.805) | 0.663 | .449 | -0.021 (-1.924–1.881) | 0.964 | **.982** |
| | Education (no = 0 / yes = 1) | | | | | | |
| | Primary school | -1.390 (-16.756–13.976) | 7.746 | .858 | -0.027 (-21.784–21.729) | 10.992 | .998 |
| | Secondary school | -1.487 (-13.239–10.265) | 5.822 | .800 | 1.084 (-15.197–17.366) | 8.103 | .894 |
| | Vocational training | -1.236 (-12.968–10.495) | 5.805 | .832 | 0.525 (-15.711–16.761) | 8.071 | .948 |
| | Highschool education | -2.748 (-15.340–9.843) | 6.249 | .662 | 3.565 (-13.217–20.347) | 8.389 | .672 |
| | University degree | -1.930 (-13.748–9.888) | 5.847 | .743 | 2.548 (-13.702–18.798) | 8.080 | .754 |
| | Master | -1.666 (-13.516–10.184) | 5.860 | .777 | 1.415 (-14.761–17.591) | 8.045 | .861 |
| | PhD | -1.129 (-12.946–10.687) | 5.875 | .848 | -1.172 (-17.540–15.197) | 8.175 | .887 |
| | Other | 0.680 (-12.108–13.469) | 6.400 | .916 | -8.076 (-25.653–9.502) | 8.840 | .364 |
| | Familiarity | 0.461 (-0.050–0.972) | 0.260 | .077 | -0.208 (-1.011–0.595) | 0.407 | .610 |
| | Previous treatment (no = 0 / yes = 1) | -0.212 (-1.389–0.965) | 0.598 | .724 | -0.723 (-2.460–1.014) | 0.883 | .414 |
| | STIGMA | | | | | | |
| | BMI | -0.011 (-0.054–0.031) | 0.021 | .604 | 0.020 (-0.041–0.080) | 0.031 | .517 |
| | **CAMI** | **0.054 (0.020–0.088)** | **0.017** | **.002**\*\* | **-0.103 (-0.155 –-0.051)** | **0.026** | **<.001**\*\*\* |
| **BLOCK 3** | DEMOGRAPHICS | | | | | | |
| | Constant | 10.011 (-5.188–25.209) | 7.495 | .190 | **41.794 (21.410–62.179)** | **10.099** | **<.001**\*\*\* |
| | **Age** | **0.066 (0.021–0.110)** | **0.023** | **.004**\*\* | -0.039 (-0.105–0.027) | 0.034 | .247 |
| | Gender (female = 0, male = 1) | -0.540 (-1.814–0.733) | 0.645 | .403 | 0.050 (-1.768–1.869) | 0.921 | .957 |
| | Education (no = 0 / yes = 1) | | | | | | |
| | Primary school | -1.208 (-17.336–14.920) | 8.085 | .882 | -0.416 (-22.926–22.093) | 11.300 | .971 |
| | Secondary school | -1.394 (-14.405–11.616) | 6.380 | .828 | 0.871 (-17.342–19.083) | 8.945 | .923 |
| | Vocational training | -2.209 (-15.232–10.813) | 6.377 | .731 | 2.420 (-15.861–20.700) | 8.964 | .789 |
| | Highschool education | -1.475 (-15.289–12.338) | 6.791 | .829 | 1.023 (-17.686–19.733) | 9.234 | .912 |
| | University degree | -1.391 (-14.488–11.706) | 6.413 | .830 | 1.452 (-16.844–19.747) | 8.974 | .873 |
| | Master | -1.142 (-14.272–11.988) | 6.427 | .860 | 0.351 (-17.845–18.546) | 8.927 | .969 |

*(Continued)*

**Table 1.** (Continued)

| N = 360 | | ATSPPHS-SF | | | SSOSH | | |
|---|---|---|---|---|---|---|---|
| | | b (95% CI) | SE B | p | b (95% CI) | SE B | p |
| | PhD | -2.071 (-15.082–10.941) | 6.407 | .748 | 0.657 (-17.510–18.825) | 8.960 | .942 |
| | Other | -0.187 (-13.896–13.522) | 6.807 | .978 | -6.397 (-24.991–12.196) | 9.268 | .493 |
| | **Familiarity** | **0.644 (0.138–1.150)** | **0.257** | **.013*** | -0.571 (-1.334–0.193) | 0.387 | .142 |
| | Previous treatment (no = 0 / yes = 1) | -0.425 (-1.581–0.730) | 0.587 | .469 | -0.301 (-1.947–1.345) | 0.836 | .719 |
| | STIGMA | | | | | | |
| | BMI | -0.016 (-0.057–0.026) | 0.021 | .454 | 0.029 (-0.028–0.086) | 0.029 | .319 |
| | **CAMI** | **0.076 (0.041–0.111)** | **0.018** | **<.001\*\*\*** | **-0.146 (-0.199 –-0.093)** | **0.027** | **<.001\*\*\*** |
| | CULTURE | | | | | | |
| | **Cuba = 0 / Germany = 1** | **-3.828 (-5.423 –-2.232)** | **0.810** | **<.001\*\*\*** | **7.562 (5.244–9.880)** | **1.176** | **<.001\*\*\*** |
| | | ATSPPHS-SF: $R^2$ = .10 for Block 1; $\Delta R^2$ = .04 for Block 2 (p = .002); $\Delta R^2$ = .06 for Block 3 (p <.001) | | | SSOSH: $R^2$ = .08 for Block 1; $\Delta R^2$ = .07 for Block 2 (p <.001); $\Delta R^2$ = .10 for Block 3 (p <.001) | | |

Reference category for education is no education at all. Significant results are printed in bold. N = sample size; b = b-value; 95% CI = 95% confidence interval; SE B = standard error of b; p = significance level; ATSPPHS-SF = Attitudes toward seeking professional psychological help—Short form [112]; BMI = Beliefs about mental illness [10]; CAMI = Community attitudes toward the mentally ill [109]; SSOSH = Self-stigma of seeking help [49].

\* p <.05;

\*\* p <.01;

\*\*\* p≤.001

measures, only age remained significant. Further, more positive beliefs about mental illness predicted less self-stigma. Again, including the stigma measures improved the model significantly, $F(2, 155)$ = 17.134, $p$ <.001, and a total of 28% of the variance of self-stigma were explained (Table 3).

## Interactions of cultural context and gender

To test interactions between gender and culture in the prediction of help-seeking attitudes, we run moderation analyses with community attitudes as apparently more relevant predictor in the context of the current investigation. First, we tested a two-way interaction between community mental health stigma and cultural context in the prediction of help-seeking attitudes. Both the direct path, $t(113.180)$ = 2.027, $p$ = .045*, and the interaction term, $t(204.106)$ = 2.536, $p$ = .012, were significant. Interestingly, this more economic model explained 16% of the variance of help-seeking attitudes (Fig 1). Next, we tested two-way and three-way interactions between cultural context and gender in the prediction of help-seeking attitudes. Only the two-way interaction was significant, $t(164.583)$ = 2.665, $p$ = .008**. Neither the direct path, $t(83.503)$ = 1.909, $p$ = .279, nor the three-way interaction were significant, $t(94.803)$ = -1.080, $p$ = .283, n.s. (Fig 2). The second moderation model explained a total of 17% of variance in help-seeking attitudes and did not improve the prediction compared to the first moderation model, $F(4, 245)$ = 0.513, $p$ = .726, n.s.

## Discussion

The results of the current study show both similarities and differences in mental health stigma and willingness to seek professional help between a Cuban and German sample. Our hypotheses were partly supported by the data. In the course of the discussion, we will comment the current findings and interpret them in the light of cultural context and the peculiarities of the

**Table 2. Multiple hierarchical regressions with two blocks of predictors on both help-seeking measures in the Cuban subsample.**

| n = 195 | | ATSPPHS-SF | | | SSOSH | | |
|---|---|---|---|---|---|---|---|
| | | **b (95% CI)** | **SE B** | **p** | **b (95% CI)** | **SE B** | **p** |
| | DEMOGRAPHICS | | | | | | |
| **BLOCK 1** | **Constant** | **21.921 (7.806–36.036)** | **6.850** | **.004**\** | 18.019 (-2.473–38.512) | 10.025 | .082 |
| | **Age** | **0.065 (0.013–0.117)** | **0.026** | **.016**\* | 0.032 (-0.047–0.111) | 0.040 | .418 |
| | Gender (female = 0, male = 1) | -0.558 (-2.514–1.398) | 0.980 | .571 | 1.041 (-1.767–3.849) | 1.407 | .462 |
| | Education (no = 0 / yes = 1) | | | | | | |
| | Primary school | -3.291 (-20.133–13.551) | 8.380 | .696 | 2.047 (-21.923–26.018) | 11.939 | .865 |
| | Secondary school | -2.342 (-15.916–11.232) | 6.590 | .725 | 2.010 (-17.515–21.715) | 9.523 | .827 |
| | Vocational training | -2.307 (-16.187–11.573) | 6.708 | .734 | 2.104 (-17.968–22.176) | 9.701 | .830 |
| | Highschool education | -1.885 (-15.983–12.213) | 6.815 | .785 | 2.087 (-18.446–22.620) | 9.919 | .835 |
| | University degree | -0.495 (-14.402–13.411) | 6.725 | .942 | -1.580 (-21.520–18.360) | 9.650 | .871 |
| | Master | -1.432 (-15.222–12.359) | 6.712 | .833 | -0.747 (-20.426–18.932) | 9.590 | .938 |
| | PhD | -0.710 (-15.093–13.673) | 7.080 | .921 | -6.641 (-26.597–13.315) | 9.857 | .505 |
| | Other | -1.075 (-17.424–15.273) | 8.002 | .894 | 1.417 (-20.984–23.818) | 11.013 | .898 |
| | Familiarity | 0.255 (-0.370–0.879) | 0.316 | .422 | -0.703 (-1.717–0.310) | 0.510 | .171 |
| | **Previous treatment** (no = 0 / yes = 1) | **-2.016 (-3.649 –-0.383)** | **0.826** | **.016**\* | 1.191 (-1.171–3.553) | 1.194 | .320 |
| | DEMOGRAPHICS | | | | | | |
| **BLOCK 2** | Constant | 13.839 (-2.955–30.633) | 8.208 | .103 | **40.720 (17.028–64.412)** | **11.544** | **.002**\** |
| | **Age** | **0.065 (0.013–0.117)** | **0.026** | **.015**\* | 0.025 (-0.053–0.102) | 0.039 | .528 |
| | Gender (female = 0, male = 1) | -0.448 (-2.398–1.502) | 0.976 | .648 | 0.835 (-1.940–3.609) | 1.384 | .549 |
| | Education (no = 0 / yes = 1) | | | | | | |
| | Primary school | -3.034 (-19.584–13.517) | 8.241 | .714 | 0.117 (-23.391–23.625) | 11.664 | .992 |
| | Secondary school | -2.029 (-15.454–11.397) | 6.519 | .758 | 1.213 (-18.798–21.224) | 9.637 | .901 |
| | Vocational training | -2.415 (-15.975–11.144) | 6.562 | .716 | 2.493 (-17.657–22.642) | 9.673 | .799 |
| | Highschool education | -2.122 (-15.850–11.605) | 6.646 | .752 | 2.705 (-17.714–23.125) | 9.807 | .785 |
| | University degree | -0.780 (-14.477–12.918) | 6.628 | .907 | -0.447 (-20.505–19.611) | 9.642 | .963 |
| | Master | -1.772 (-15.335–11.791) | 6.605 | .791 | 0.354 (-19.463–20.171) | 9.591 | .971 |
| | PhD | -0.778 (-14.781–13.225) | 6.903 | .911 | -6.002 (-25.552–13.549) | 9.618 | .537 |
| | Other | -0.700 (-17.424–16.023) | 8.161 | .932 | -0.042 (-23.095–23.010) | 11.233 | .997 |
| | Familiarity | 0.152 (-0.474–0.777) | 0.317 | .633 | -0.396 (-1.341–0.549) | 0.476 | .407 |
| | **Previous treatment** (no = 0 / yes = 1) | **-1.867 (-3.476 –-0.257)** | **0.814** | **.023**\* | 0.801 (-1.427–3.028) | 1.125 | .478 |
| | STIGMA | | | | | | |
| | BMI | 0.026 (-0.024–0.076) | 0.025 | .297 | -0.028 (-0.094–0.039) | 0.034 | .415 |
| | **CAMI** | **0.048 (0.000–0.096)** | **0.024** | **.050**\* | -0.148 (-0.226 –-0.070) | 0.039 | <.001\*** |
| | | ATSPPHS-SF: $R^2$ = .12 for Block 1; $\Delta R^2$ = .03 for Block 2 (p = .098, n.s.) | | | SSOSH: $R^2$ = .11 for Block 1; $\Delta R^2$ = .12 for Block 2 (p ≤ .001) | | |

Reference category for education is no education at all. Significant results are printed in bold. n = size of the subsample; b = b-value; 95% CI = 95% confidence interval; SE B = standard error of b; p = significance level; ATSPPHS-SF = Attitudes toward seeking professional psychological help—Short form [112]; BMI = Beliefs about mental illness [10]; CAMI = Community attitudes toward the mentally ill [109]; SSOSH = Self-stigma of seeking help [49].

\* p <.05;

\** p <.01;

\*** p ≤.001

current samples. The overall level of mental health stigma was moderate to low in both samples, but positive help-seeking attitudes were not as strong as desirable. Although German samples showed moderate to adverse attitudes towards mental illness compared to other countries before [129], Cubans show even more mental health stigma. This finding fits well with the

**Table 3. Multiple hierarchical regressions with two blocks of predictors on both help-seeking measures in the German subsample.**

| n = 165 | | ATSPPHS | | | SSOSH | | |
|---|---|---|---|---|---|---|---|
| | | b (95% CI) | SE B | p | b (95% CI) | SE B | p |
| **BLOCK 1** | DEMOGRAPHICS | | | | | | |
| | **Constant** | **15.350 (10.960–20.890)** | **2.666** | **<.001*** | **35.183 (33.440–44.540)** | **4.160** | **<.001*** |
| | **Age** | | 0.042 | .874 | **-0.172 (-0.304 – -0.040)** | **0.066** | **.010*** |
| | Gender (female = 0, male = 1) | 0.007 (-0.068–0.092) | 0.843 | .442 | 0.110 (-2.351–2.915) | 1.315 | .934 |
| | Education (no 0 / yes = 1) | -0.650 (-2.643–0.0314) | | | | | |
| | Highschool education | | 1.989 | .732 | -2.156 (-11.401–1.981) | 3.103 | .488 |
| | University degree | | 1.890 | .796 | -1.419 (-9.128–3.100) | 2.949 | .631 |
| | Other | -0.684 (-4.952–2.347) | 2.868 | .744 | -1.581 (-12.128–5.733) | 4.476 | .724 |
| | **Familiarity** | -0.489 (-4.460–2.672) | **0.423** | **<.001*** | **-1.424 (-2.699–0.043)** | **0.660** | **.033*** |
| | **Previous treatment** | -0.938 (-7.449–3.531) | **0.859** | **.049*** | -2.352 (-4.220–0.835) | 1.341 | .081 |
| | (no = 0 / yes = 1) | **1.983 (1.063–2.739)** | | | | | |
| | | **1.707 (-0.009–3.419)** | | | | | |
| **BLOCK 2** | DEMOGRAPHICS | | | | | | |
| | **Constant** | 5.609 (-6.405–19.019) | 6.238 | .370 | **35.643 (22.18–66.86)** | **9.541** | **<.001*** |
| | **Age** | | 0.040 | .651 | **-0.197 (-0.317 – -0.027)** | **0.060** | **.001**** |
| | Gender (female = 0, male = 1) | 0.018 (-0.038–0.103) | 0.793 | .920 | -0.652 (-3.856–1.147) | 1.213 | .592 |
| | Education (no 0 / yes = 1) | -0.080 (-1.717–1.283) | | | | | |
| | Highschool education | -1.330 (-6.127–0.845) | 1.853 | .474 | -0.897 (-7.188–3.313) | 2.835 | .752 |
| | University degree | -0.651 (-4.300–2.202) | 1.757 | .712 | -1.065 (-7.875–2.985) | 2.687 | .692 |
| | Other | -1.063 (-8.030–2.765) | 2.666 | .691 | -1.519 (-9.859–5.656) | 4.077 | .710 |
| | **Familiarity** | **1.519 (0.488–2.265)** | **0.413** | **<.001*** | -0.403 (-1.732–1.231) | 0.632 | .524 |
| | Previous treatment (no = 0 / yes = 1) | 0.634 (-0.911–2.321) | 0.830 | .446 | -0.329 (-2.249–3.176) | 1.269 | .796 |
| | STIGMA | | | | | | |
| | **BMI** | -0.053 (-0.141–0.016) | 0.039 | .172 | **0.193 (0.090–0.317)** | **0.059** | **.001**** |
| | **CAMI** | **0.078 (0.023–0.134)** | **0.027** | **.005**** | -0.063 (-0.155–0.017) | 0.041 | .129 |
| | | ATSPPHS-SF: $R^2$ = .20 for Block 1; $\Delta R^2$ = .12 for Block 2 ($p \leq .001$) | | | SSOSH: $R^2$ = .12 for Block 1; $\Delta R^2$ = .16 for Block 2 ($p \leq .001$) | | |

Reference category for education is no education at all. Significant results are printed in bold. n = size of the subsample; b = b-value; 95% CI = 95% bootstrapped confidence interval; SE B = standard error of b; p = significance level; ATSPPHS-SF = Attitudes toward seeking professional psychological help—Short form [112]; BMI = Beliefs about mental illness [10]; CAMI = Community attitudes toward the mentally ill [109]; SSOSH = Self-stigma of seeking help [49].

* $p < .05$;

** $p < .01$;

*** $p \leq .001$

generally high mental health stigma found in US-Latin American [51, 55, 56, 64] and US-Cuban samples [61]. High mental health stigma in Latin American populations is often explained by explanatory models which associate mental illness with shame and weakness [61] or embarrassment [105]. Yet, findings are mixed and a compelling theory is still lacking. For example, Peluso and Blay [130] report predominantly positive attitudes about mental illness in a review of studies from different Latin American countries. These heterogeneous results might be explained by the diverse research settings and samples which complicate direct comparisons. Regarding willingness to seek professional help, we found more negative attitudes and more self-stigma in the German compared to the Cuban sample. This finding matches some evidence of comparably negative help-seeking attitudes in Germany compared to other European countries [113, 131]. However, previous evidence again is mixed and reports both a

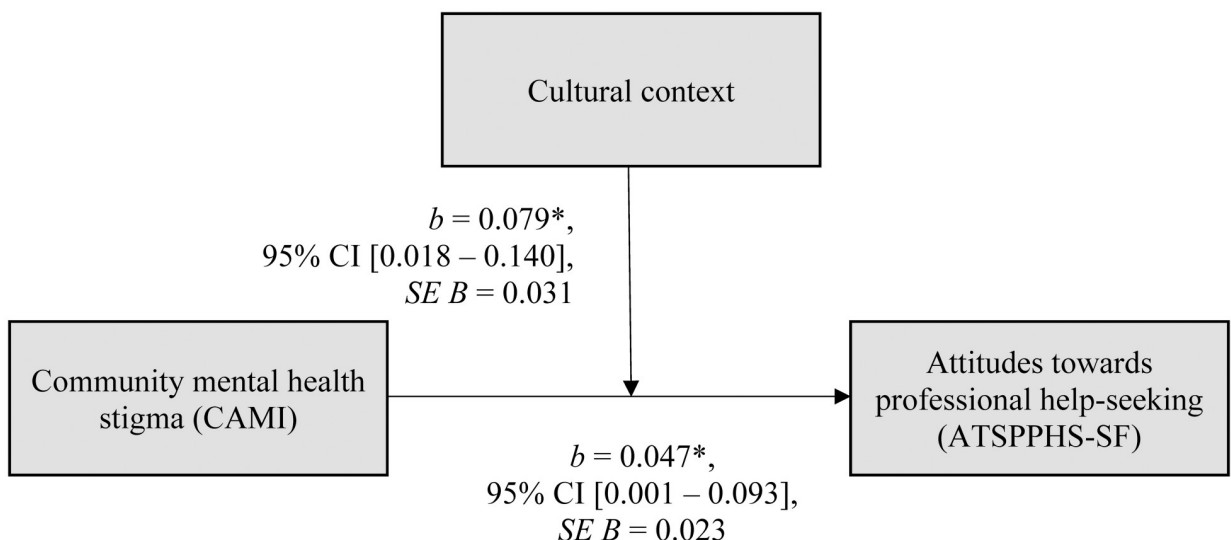

**Fig 1. Moderation model.** $R^2$ = .16. Cultural context: Cubans = 0, Germans = 1. $b$ = $b$-value; 95% CI = 95% confidence interval; $t$ = $t$-statistic; $p$ = significance level; ATSPPHS-SF = Attitudes toward seeking professional psychological help—Short form [112]; CAMI = Community attitudes toward the mentally ill [109]. * $p$ <.05; ** $p$ <.01; *** $p$≤.001.

comparable level of self-stigma [123] and less self-stigma on help-seeking in German samples [125]. Still, when interpreting this finding one has to keep in mind the preexisting differences between the Cuban and German samples. Especially the role of age on mental health stigma and help-seeking attitudes remains vague in the literature. Assuming that higher age is associated with more mental health stigma and less openness to professional help [e.g., 113, 132], the positive help-seeking attitudes found in the Cuban sample would impress even more. Yet, we identified a trend of more positive attitudes and less self-stigma with older age in both cultural

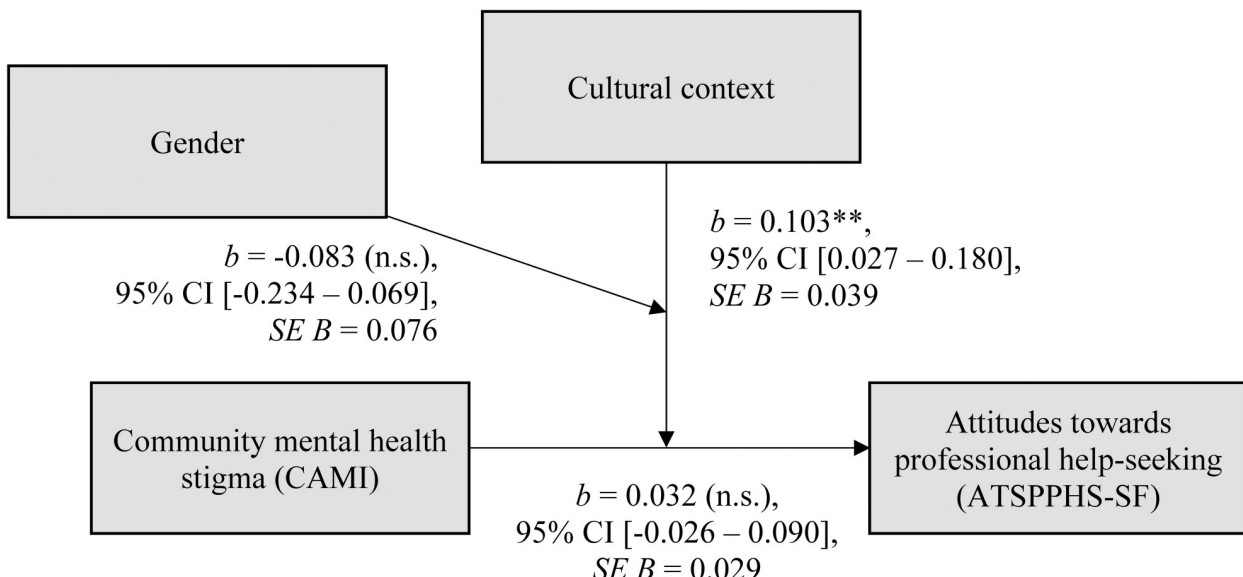

**Fig 2. Moderated moderation model.** $R^2$ = .17. Cultural context: Cubans = 0, Germans = 1. Gender: female = 0, male = 1. $b$ = $b$-value; 95% CI = 95% confidence interval; $t$ = $t$-statistic; $p$ = significance level; ATSPPHS-SF = Attitudes toward seeking professional psychological help—Short form [112]; CAMI = Community attitudes toward the mentally ill [109]. * $p$ <.05; ** $p$ <.01; *** $p$≤.001.

groups (see Tables 2 and 3) and current literature hints in the same direction [e.g., 133, 134], which might challenge the role of culture in this finding at least partly. Thus, it seems especially important to examine the effect of age on help-seeking attitudes and potential interactions with the cultural context in the future.

As predicted by our first hypothesis, mental health stigma was found to be a universal mental health care barrier. In both samples negative community attitudes predicted negative help-seeking attitudes. Thus, the way people anticipate how society will treat them when labeled as mentally ill seems to inhibit help-seeking regardless of the cultural context. The reduction of social discrimination associated with mental illness therefore remains a main goal to improve mental health care use. Following this idea, we found an interesting interaction of mental health stigma and help-seeking attitudes between Cuba and Germany. Despite higher mental health stigma, Cubans reported higher willingness to seek professional help which is against our second hypothesis. Ojeda and Bergstresser [44] conceptualize *illness danger* as the amount of social status one may lose due to mental illness and professional help-seeking. Thus, Cuban participants may perceive less illness danger due to the special social setting: Generally, (mental) illness is said to be no risk factor for poverty in Cuba [71]. Beyond, stigma at work is an often-mentioned burden of mentally ill persons which well fits the German cultural context of professional success being considered a highly self-esteem relevant aspect of everyday life. As opposed to this, in the Cuban socialist economy all citizens are meant to work and to support the society, and even severe mentally ill persons are compelled to work [76, 135]. In this way, they remain integrated in work life and may be able to maintain their self-esteem. At this point, the age difference of the samples again might have influenced the findings. Since German participants were significantly younger than the Cubans one might argue that mental health stigma hampers help-seeking intentions in the German sample only based on the younger age [38]. Still, older adults may also perceive less illness danger compared to younger adults which might be afraid of losing their future or their role in society due to help-seeking. This idea merits further investigation since it might inform powerful anti-stigma interventions and strategies to facilitate mental health care access. Nevertheless, subgroup analyses show more culture specific effects. In the Cuban sample, community attitudes were more relevant in the prediction of willingness to seek help compared to individual beliefs which matches well the socio-centric personal orientation of the Cuban culture [56, 79, 80]. Correspondingly, mental health stigma is conceptualized as a community-level barrier to mental health care in Latin American samples [55, 56]. In the German sample, findings are more nuanced: Individual beliefs predicted self-stigma while community attitudes predicted help-seeking attitudes. Thus, label avoidance by avoiding help-seeking might be particularly important to Germans with especially negative beliefs about mental illness to protect their self-esteem [136]. Moreover, in more individualistic cultures personal decisions and independence are important values [86] and handling emotional problems without help might be a significant strategy to protect self-esteem.

Due to the mixed evidence regarding gender and help-seeking [44, 51, 137–139] and the importance of gender roles in Latin American cultures [56], we investigated its influence exploratory as stated by our third hypothesis. No significant effect was found and the interaction terms did not improve the prediction. Further, gender was no significant predictor in the regression models. This finding could be due to the fact that gender roles are less pronounced in Cuba and Germany. Both countries share a history of socialist politics where women are not only dedicated to motherhood and housekeeping, but also supposed to work full time. For example, Cuban women usually return to work after twelve months of parental leave. This might be one reason why marianismo is not equally strong in Cuba compared to other Latin American cultures. Still, men seek less professional help across cultures [44, 139] which is

generally explained by conformity to masculinity norms [140] or *machismo* in the case of Latin American men [141]. One probable explanation of why we did not replicate this finding may be the fact that we only assessed gender, not identification with gender roles.

Beyond our hypotheses, we found interesting results regarding the included social demographics and control variables. As mentioned above, older age predicted more positive help-seeking attitudes in the Cuban sample while it predicted less self-stigma on help-seeking in the German sample. Generally, older (vs. younger) adults seek less professional help. Literature discusses negative attitudes, less mental health literacy, and a lack of health insurance for older adults as relevant causes [132 for an overview]. At the same time, studies show comparable results between younger and older adults [107, 132] or suggest even more positive attitudes in older adults [44, 133]. In the study of Rojas-Vilches et al. [61], older Cubans reported more negative beliefs about mental illness and less willingness to seek professional help. Positive help-seeking attitudes of older Cubans living in Cuba may be attributable to the massive development of the Cuban health care system. In this sense, older generations might appreciate the possibility to seek professional help without charge when in need which might be reflected in more positive help-seeking attitudes compared to younger Cubans. This might further explain why only help-seeking attitudes were associated with age, not self-stigma on help-seeking. In the German sample, older age was predictive of less self-stigma on help-seeking which does not fit previous findings from Germany [113]. This might be explained by the fact that self-esteem is less stable in younger ages in general [142] and might by stronger effected by group identification and public stigma [38]. Still, when interpreting these findings, one has to consider the young mean age and the low variance of age in the German sample. As discussed above, the differences in mean age and variance between the two samples might have biased the results.

More familiarity with mental illness only predicted positive help-seeking attitudes in the German sample. One common explanation of this finding is the contact hypothesis which postulates that contact between groups reduces negative reactions due to familiarization [143]. Thus, being familiar with mentally ill persons might reduce label avoidance which in turn might reduce negative help-seeking attitudes. Hereof, one would expect a higher level of familiarity in the Cuban sample considering the social psychiatric approach of the mental health care system [73, 144, 145]. Problematically, we included familiarity with mental illness only as a one item control variable and did not use a more nuanced measure [e.g., 8] to capture this concept. Moreover, as our questionnaires did not specify 'mental illness', Cuban participants might have not considered mentally ill relatives while answering the survey since they might label them differently. Label avoidance by calling affected relatives *enfermo/a de los nervios* instead of using the more stigmatizing label *loco/a* seems to be a relevant strategy to handle mental health stigma in Latin American cultures [55, 56]. This idea is supported by the current finding that previous help-seeking did not predict self-stigma on help-seeking. Hence, affected Cubans might not internalize mental health stigma after seeking professional help because of not identifying with the group of mentally ill persons. Still, former help-seeking predicted considerably negative help-seeking attitudes in the Cuban sample. Negative experiences in the past seem an unlikely explanation of this finding because of the relatively high rating of helpfulness of previous treatment. Thus, the current data does not provide a satisfying explanation for this finding and more research is needed. Particularly, future research should investigate familiarity with mental illness and the experiences of previous professional help-seeking in the Cuban context more differentiated to understand their influence in mental health stigma and help-seeking behavior more rigorously.

## Limitations and future directions

This study has several limitations. First, both samples are not representative for the respective society because of opportunity sampling. Especially the German sample is much younger and more educated than the German average. Interestingly, although education is a frequently discussed factor influencing mental health stigma, it was of little importance in both cultural samples in the current study. Especially problematic seems the considerably higher mean age in the Cuban sample. Although our first regression model shows an effect of culture on both help-seeking attitudes and self-stigma on help-seeking when controlling for age (see Table 1), we cannot completely separate the effect of culture and age on the findings. Future research is needed to substantiate our results and explore the research questions raised above. Moreover, the different sampling procedures might have affected the current findings. The online sampling in the German cultural context might have attracted already interested or affected Germans. In the Cuban cultural context, the applied sampling strategy might have induced a slight bias towards a more positive help-seeking attitude because of sampling in a hospital's waiting areas. Thus, generalization of the results is not possible and findings should be interpreted as first insight in a previously understudied research context in international research. Second, the methodological differences in data collection (pencil-paper vs. online) might have biased the data and its analyses [105]. Yet, we decided to use different survey methods to reach a high fit between survey method and cultural context and to avoid the common mistake of a *one size fits all*-strategy in cross-cultural research. Additionally, our data faces a well-known methodological difficulty of quantitative cross-cultural research: The translated questionnaires show better methodological properties in the Western compared to the non-Western sample. Thus, analyses might be affected by the higher validity of assessment in the German related to the Cuban sample. To meet a minimum of reliability anyhow, we selected the used measurements carefully based on previous research and described our methods in detail. Generally, future research might therefore pursue diverse methodological approaches in the field of (mental) health behavior, for example, more descriptive, qualitative, and mixed-methods studies [e.g., 146, 147]. Besides the mentioned limitations, one strength of the current study is that we investigated attitudes of persons in their home country. Thus, mistrust and fear of the system, language barriers, coverage of health insurance, or other important barriers found in immigrant populations did not influence the results [e.g., 44]. Summarized, the current findings highlight the potential of research on mental health stigma in the Cuban cultural context. Since the comparably high mental health stigma is not likewise reflected in more negative help-seeking attitudes, it might be helpful to understand how help-seeking behavior can be improved besides (high) mental health stigma.

The current study offers first insights in mental health stigma and professional help-seeking attitudes in the Cuban cultural context. Further, it shows considerable similarities and differences between Cuba and Germany. The postulated models of the current study explain at most 32% of variance in help-seeking attitudes and self-stigma on help-seeking. Although this is not even one third of the variance, these models offer crucial information on how to promote mental health care use differently for Cuban and German general population as well as for different ages. Thus, it seems more important to strengthen the individual self-esteem regarding mental health and help-seeking in Germans and younger persons–maybe using familiarization. While it seems more crucial to address the strong mental health stigma at community level in the Cuban general population and older adults using popular media like television or fotonovelas as proposed by Cabassa and colleagues [e.g., 148, 149]. Still, other culturally informed aspects might influence the help-seeking process. In this sense, explanatory models, social support, and informal sources of help like spirituality, religion, family, and friends

should be considered in future research [52, 53, 55, 56, 61, 76, 130]. We hope that the current study will inspire more cross-cultural research on mental health stigma and help-seeking. In sum, the current study adds further evidence to the literature on mental health stigma and help-seeking attitudes. Consequently, this may support future reviews and meta-analyses to promote evidence-based and culturally informed interventions in view of improving mental health care globally.

## Supporting information

**S1 Table. Absolute and relative frequencies of female gender, level of education, and previous help-seeking in the Cuban and German samples.** *n* = size of the subsample; * statistically significant after Bonferroni correction (*p* <.005).
(TIF)

**S2 Table. Mean and standard deviation of age, familiarity with mental illness, and helpfulness of previous help-seeking in the Cuban and German samples.** *n* = size of the subsample; *SD* = standard deviation; * statistically significant after Bonferroni correction (*p* <.005).
(TIF)

**S3 Table. Mean and standard deviation of mental health stigma measures and attitudes regarding professional help-seeking in the Cuban and German sample.** *n* = sample size of subsample; *SD* = standard deviation; ATSPPHS-SF = Attitudes toward seeking professional psychological help—Short form [112]; BMI = Beliefs about mental illness [10]; CAMI = Community attitudes toward the mentally ill [109]; SSOSH = Self-stigma of seeking help [49]; * statistically significant after Bonferroni correction (*p* <.005).
(TIF)

## Acknowledgments

We thank Susell F. Alvarez Castañeda for her assistance in data collection and translation of the questionnaires and Christina Hanna for her assistance in translation of the questionnaires, data collection, and programming.

## Author Contributions

**Conceptualization:** Laura Nohr, Alexis Lorenzo Ruiz, Juan E. Sandoval Ferrer, Ulrike Buhlmann.

**Data curation:** Laura Nohr.

**Formal analysis:** Laura Nohr.

**Funding acquisition:** Laura Nohr.

**Investigation:** Laura Nohr.

**Methodology:** Laura Nohr, Ulrike Buhlmann.

**Project administration:** Laura Nohr.

**Resources:** Ulrike Buhlmann.

**Supervision:** Alexis Lorenzo Ruiz, Juan E. Sandoval Ferrer, Ulrike Buhlmann.

**Validation:** Alexis Lorenzo Ruiz, Juan E. Sandoval Ferrer, Ulrike Buhlmann.

**Writing – original draft:** Laura Nohr.

**Writing – review & editing:** Alexis Lorenzo Ruiz, Juan E. Sandoval Ferrer, Ulrike Buhlmann.

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
