## [Decision Letter · Decision Letter 0]

1 Dec 2020

PONE-D-20-28889

Mental health stigma and professional help-seeking attitudes - A comparison between Cuba and Germany

PLOS ONE

Dear Dr. Buhlmann,

Thank you for submitting your manuscript to PLOS ONE. After careful consideration, we feel that it has merit but does not fully meet PLOS ONE’s publication criteria as it currently stands. Therefore, we invite you to submit a revised version of the manuscript that addresses the points raised during the review process.

We look forward to receiving your revised manuscript.

Kind regards,

Ricarda Nater-Mewes, PhD

Academic Editor

PLOS ONE

Journal Requirements:

2. Please ensure that you include a title page within your main document.

We do appreciate that you have a title page document uploaded as a separate file, however, as per our author guidelines (http://journals.plos.org/plosone/s/submission-guidelines#loc-title-page) we do require this to be part of the manuscript file itself and not uploaded separately.

4. Please ensure that you refer to Figure 2 in your text as, if accepted, production will need this reference to link the reader to the figure.

Additional Editor Comments:

The Reviewers and the Editor see much merit in the manuscript and consider its topic as being very relevant. However, some limitations remain, mainly with regard to the sampling procedure and the statistical anlayses, which should be addressed in a revised version.

Reviewers' comments:

Reviewer's Responses to Questions

**Comments to the Author**

1. Is the manuscript technically sound, and do the data support the conclusions?

Reviewer #1: Yes

Reviewer #2: No

2. Has the statistical analysis been performed appropriately and rigorously? 

Reviewer #1: Yes

Reviewer #2: No

3. Have the authors made all data underlying the findings in their manuscript fully available?

Reviewer #1: No

Reviewer #2: No

4. Is the manuscript presented in an intelligible fashion and written in standard English?

Reviewer #1: Yes

Reviewer #2: Yes

5. Review Comments to the Author

Reviewer #1: COMMENTS.

This is a well written manuscript that focuses on the important subject matter of mental health stigma.

1. There is a need for a bit more description of the sampling method. It clearly is a sample of convenience/ purposive sample. Kindly give a bit more description of regions it surveyed and how the survey was spread in thes 2 settings. I am however glad that this is highlighted as a limitation at the end of the manuscript.

2. The important aspect of ethical considerations was however left out in the manuscript proper. I would like the authors to include a paragraph on ethical considerations just before the results section. Issues on Ethical approval, consent/autonomy, confidentiality, anonymity, beneficience/maleficience should all be duely highlighted as needed.

The manuscript may be considered for publication by editor once these have been done

Reviewer #2: Thank you for the opportunity to review this paper on mental health stigma and professional help-seeking attitudes. It's an important topic, given the proportion of the population with common mental health disorders. This study is a cross-sectional survey of Cubans (n = 195) and Germans (n = 165) adult samples.

The introduction considers relevant literature and is well structured. The rationale for the study could be strengthened, and the importance of cross-cultural research outlined in a bit more detail.

It’s good that the authors aimed to sample the general population in both cultural contexts. However, the sample recruitment strategy is very different and the sample characteristics reflect this. The Cuban sample was recruited from hospital waiting rooms, which may imply poor health status and a general willingness to seek help. The German sample was recruited into an online survey through social media, newspapers, bus stations and supermarket. There are issues with generalisability to the population (explicitly mentioned for the German sample) but also issues when comparing the two samples against each other. Can the authors discuss to the extent they think differences in help-seeking between samples is a function of age and recruitment strategy, rather than real differences between Cubans and Germans? The authors allude to this in lines 446-447, but this needs more explicit and fuller consideration in the limitations section.

Missing data from Cuban sample – Little’s test suggests NMCAR, so not clear what subsequent analyses were conducted to suggest missing at random? I would encourage authors to conduct analyses both with the imputed data (as presented here) and also the conduct the analyses on the completed responses to see how results differ.

Results

The authors conduct t-tests on categorical data (education, gender, previous help-seeking), for which I would have expected chi-square analyses. Can the authors comment on the choice of analysis for the categorical variables?

It's not clear from the description what reference categories have been used for categorical demographic variables in the regression analysis; include this additional detail in the regression tables.

I would suggest re-running the first regression using dummy codes 0-1 rather than 1-2 for country, as the use of 1-2 impedes interpretability of the regression analysis for the reader.

Table with all mean values and possible values to allow for interpretation of sample averages

Discussion

Further consideration of how the knowledge from this study can inform culturally informed interventions would be a good addition.

6. PLOS authors have the option to publish the peer review history of their article (what does this mean?). If published, this will include your full peer review and any attached files.

Reviewer #1: No

Reviewer #2: No

---

## [Author Response · Author response to Decision Letter 0]

23 Dec 2020

Dear Dr. Nater-Mewes, dear Reviewers, 

We appreciate your helpful comments and considerations on our manuscript about mental health stigma in Cuba and Germany. Below, we address each of your comments carefully and explain the way we implemented them in the revised manuscript. We would like to address the Editors comments first:

1. We checked the PLOS ONE’s style requirements and hope that the manuscript and all related files correspond to them now. If we still miss important requirements, please let us know. 

2. Thank you for the gentle hint about the title page. We are sorry about having misinterpreted the PLOS ONE author guidelines before. We included the title page in the manuscript as required. 

3. As required, we added additional information about our data sharing policy in the Cover Letter. We completely support Open Science in psychological research and will attempt to meet all the necessary requirements from the beginning of current and future research projects. Unfortunately, due to ethical requirements and after communicating with the Institutional Review Board we will not be able to make data openly available. In the Cover letter we describe in detail, what might be a solution in this certain case. If we left any open issues, please let us know. We are strongly willing to cooperate as much as possible within the ethical possibilities. 

4. We appreciate the useful hint about the figures. We referred erroneously two times two figure 1 instead of one time to each figure. We corrected this mistake in the revised manuscript. 

Now we would like to answer the helpful and thoughtful comments of Reviewer 1. Reviewer 1 states that the topic of mental health stigma is important and we appreciate that she/ he has given us the chance to improve our manuscript to share our findings with the PLOS ONE community in the future. 

1. We comprehend completely the wish of further information regarding the sampling procedure in both cultural contexts. We added the required information in the methods section and hope to clarify the convenient nature of our sample which only allows us to share explorative findings. We are completely aware of the fact that those findings cannot be generalized and need further validation in another, preferential more representative sample. 

2. We are glad to have the chance to add this value information on ethical considerations. Instead of the short sentences we included in the first manuscript, we added a whole paragraph to describe the ethical considerations underlying this study in more detail. 

In the following, we would like to address the important issues raised by Reviewer 2. We think that she/ he addressed precisely the main weaknesses of the current manuscript. Still, we are convinced about the importance of our findings although we cannot offer generalizability of the results. 

1. Thank you for your positive feedback on our introduction. As proposed, we added some information to strengthen the rationale of cross-cultural research in general and of the study itself. We think cross-cultural research is a necessary condition for a profound understanding of psychological phenomena and for powerful interventions in diverse populations like most Western populations in the globalized world. Thus, we need to know to which extent we are dealing with universal phenomena or culturally informed peculiarities. We hope that we strengthened this point sufficiently without broaden the introduction to much. 

2. You are absolutely right that due to preexisting differences between the samples the findings are not solely based on differences between the cultures. We addressed this point by discussing alternative interpretations in more detail, especially the possible influence of the significant difference in mean age. Further, we added some thoughts to the limitations section. Thereby, we describe possible bias due to different sampling procedures in more detail. We think that both procedures had their strength and weakness and might have biased the sampling. Still, we think that we chose the best fitting procedure for each cultural context and assume to would have introduced an even more severe bias when standardizing the procedures. We are aware that these are difficulties of cross-cultural psychological research and that we could not meet all requirements perfectly in this exploratory study. Yet, we think that the manuscript offers useful information for the scientific community interested in cross-cultural research in the Cuban cultural context and hope to inspire further research which might address methodological difficulties differently. 

3. Thank you for raising this important question. We reported the result of Little’s MCAR test since it is a well-known test and we expected most of the readership to be familiar with this test. Still, important researches in the field of missing data and multiple imputation doubt the usefulness of the test as the only measure of missing mechanism since it only offers an omnibus test of the MCAR mechanism (e.g. p. 19-21, Enders, 2010). As we aimed to use multiple imputation to handle the missing data in the Cuban data set, we used t-tests and correlation analyses to identify auxiliary variables for the imputation model (see Enders, 2010, 2016). This procedure helps to substantiate the result of Little’s test. Further, it offers hints whether the missing data is MAR or MNAR. You are absolutely right that there is no chance to differentiate entirely between MAR and MNAR. Still, as we were able to identify several variables within the data which were substantially correlated (r ≥ |.40|) with the missingness of other variables, we expect our data not to be completely MNAR. With the help of the auxiliary variables we were able to impute the missing data. Convergence and graphic diagnostic did not disturb us so that we trust the findings based on the imputed and pooled results. To reduce confusion and a lack of clarity anyway, we slightly adapted our wording regarding the missing mechanism. 

We appreciate your recommendation to apply all analyses on the complete Cuban data as well. When applying listwise deletion instead of multiple imputation, one has to keep in mind the loss of power and the difficulty of not meeting the statistical assumptions of the analyses any more: The sample seize reduces as low as to Ncompl = 73 when using listwise deletion instead of multiple imputation to handle missing data. Thus, we trust strongly more the findings from the multiple imputed data set than the results based on complete data as we expect even more bias and less power in the complete data. Still, we repeated the analyses with complete data only and were able to replicate most of the findings, but not all of them. Not every significant predictor in the regression analyses remained significant. For example, the BMI sum score becomes a significant predictor of the ATSPPHS sum score in the Cuban sample and age remains significant in the third block predicting the SSOSH sum score in the total sample. In sum, we think the findings from the multiple imputed data are much more trustworthy and do not believe that not replicating every predictor impairs the reliability of the reported findings. In sum, we would like to maintain the results and findings as presented in the manuscript. 

4. We appreciate this hint very much. It seems that we missed to adapt the R script adequately. We corrected the analyses using Chi2-tests for all categoric variables and changed the manuscript respectively. When using Chi2-tests on multiple imputed data the pooled results are using a F-statistic. The findings did not change at all by doing so. 

5. We appreciate this hint on the readability of our regression tables. We added the information requested in the tables. Regarding the dummy coded variable education, no education at all was the reference category. Although difficult to imagine from a Western perspective, especially Cuban elderly persons might not have received any formal education before the Cuban Revolution and its reform of the educational system. Thus, we wanted to be able to depict this possibility as well, when necessary.

6. Thank you for this suggestion. We re-run all regression analyses with the coding Cuba = 0 and Germany = 1 as well as female = 0 and male = 1 to enable interpretation of the results. We adapted all parameters in the tables, figures, and text which changed due to the new coding. 

7. To be honest, we included and excluded tables showing the social demographics of Cuban and German participants several times while preparing the manuscript. We agree that respective tables help the readership to gain a fast overview over the samples and to compare the main characteristics. At the same time, we are afraid of even more tables which might confuse more than it might help. In the end, this is why we decided not to include them. Moreover, we tried to avoid repetition of information within the text and the tables. Still, we think it might be helpful to include these tables in the online supplement material. For this reason, we uploaded the tables as additional files. Yet, we will include these tables in the manuscript if strongly suggested. 

8. We agree that basic knowledge gained from cross-cultural studies like the current study are very important to inform future interventions. Still, we think that the data base offered by this study does not allow to deduce concrete ideas for culturally informed interventions. We felt supported in this idea by your thoughtful reviews which hinted to the lack of generalizable findings due to the sampling procedure. Anyway, to strengthen the idea of culturally informed interventions for the future we added some cautious ideas derived from the current findings without overinterpreting them or exceeding the scope of the study. We hope that you agree with this approach and are satisfied by our implementation in the manuscript. 

Conclusive, we would like to thank you for the efforts and thoughtful comments on our manuscript. With your help, our manuscript has improved a lot. We hope that we could address your concerns adequately and that you might consider the manuscript for publication. 

Sincerely,

Prof. Dr. Ulrike Buhlmann and her co-authors

---

## [Decision Letter · Decision Letter 1]

21 Jan 2021

Mental health stigma and professional help-seeking attitudes - A comparison between Cuba and Germany

PONE-D-20-28889R1

Dear Dr. Buhlmann,

We’re pleased to inform you that your manuscript has been judged scientifically suitable for publication and will be formally accepted for publication once it meets all outstanding technical requirements.

Kind regards,

Ricarda Nater-Mewes, PhD

Academic Editor

PLOS ONE

Additional Editor Comments (optional):

Reviewers' comments:

Reviewer's Responses to Questions

**Comments to the Author**

1. If the authors have adequately addressed your comments raised in a previous round of review and you feel that this manuscript is now acceptable for publication, you may indicate that here to bypass the “Comments to the Author” section, enter your conflict of interest statement in the “Confidential to Editor” section, and submit your "Accept" recommendation.

Reviewer #2: All comments have been addressed

2. Is the manuscript technically sound, and do the data support the conclusions?

Reviewer #2: Yes

3. Has the statistical analysis been performed appropriately and rigorously? 

Reviewer #2: Yes

4. Have the authors made all data underlying the findings in their manuscript fully available?

Reviewer #2: No

5. Is the manuscript presented in an intelligible fashion and written in standard English?

Reviewer #2: Yes

6. Review Comments to the Author

Reviewer #2: Thank you for the opportunity to review the resubmission of this manuscript. The authors have addressed all my concerns and I am happy to suggest to the Editor the revised manuscript is accepted for publication.

7. PLOS authors have the option to publish the peer review history of their article (what does this mean?). If published, this will include your full peer review and any attached files.

Reviewer #2: No

---

## [Editor Report · Acceptance letter]

2 Feb 2021

PONE-D-20-28889R1 

Mental health stigma and professional help-seeking attitudes
A comparison between Cuba and Germany 

Dear Dr. Buhlmann:

I'm pleased to inform you that your manuscript has been deemed suitable for publication in PLOS ONE. Congratulations! Your manuscript is now with our production department. 

Kind regards, 

on behalf of

Dr. Ricarda Nater-Mewes 

Academic Editor

PLOS ONE